# Identification of the Tumor Infiltrating Lymphocytes (TILs) Landscape in Pure Squamous Cell Carcinoma of the Bladder

**DOI:** 10.3390/cancers14163999

**Published:** 2022-08-18

**Authors:** Lennert Eismann, Severin Rodler, Alexander Buchner, Gerald Bastian Schulz, Yannic Volz, Robert Bischoff, Benedikt Ebner, Thilo Westhofen, Jozefina Casuscelli, Raphaela Waidelich, Christian Stief, Boris Schlenker, Stephan Ledderose

**Affiliations:** 1Department of Urology, University Hospital Munich, Ludwig-Maximilian-University, 81377 Munich, Germany; 2Department of Pathology, University Hospital Munich, Ludwig-Maximilian-University, 80337 Munich, Germany

**Keywords:** bladder cancer, tumor-infiltrating lymphocytes, squamous cell carcinoma, prognosis

## Abstract

**Simple Summary:**

Treatment options in squamous cell carcinoma (SCC) of the bladder are limited and prognosis is poor. In this report we investigated the impact of tumor-infiltrating lymphocytes (TILs) in SCC of the bladder in patients undergoing radical cystectomy. We found that subsets of TILs hold predictive value for OS and PFS. We conclude that TILs might stratify patients with bladder SCC for immunotherapy.

**Abstract:**

**Background:** Tumor infiltrating lymphocytes (TILs) are known as important prognostic biomarkers and build the fundament for immunotherapy. However, the presence of TILs and its impact on outcome in pure squamous cell carcinoma (SCC) of the bladder remains uncertain. **Methods:** Out of 1600 patients undergoing radical cystectomy, 61 patients revealed pure bladder SCC in the final histopathological specimen. Retrospectively, immunohistochemical staining was performed on a subset of TILs (CD3+, CD4+, CD8+, CD20+). Endpoints were overall survival (OS), cancer-specific survival (CSS) and progression-free survival (PFS). The Kaplan–Meier method was used to evaluate survival outcomes. **Results:** Strong infiltration of CD3+ was found in 27 (44%); of CD4+ in 28 (46%); of CD8+ in 26 (43%); and of CD20+ in 27 tumors (44%). Improved OS was observed for strong CD3+ (*p* < 0.001); CD4+ (*p* = 0.045); CD8+ (*p* = 0.001); and CD20+ infiltration (*p* < 0.001). Increased rates of PFS were observed for CD3+ (*p* = 0.025) and CD20+ TILs (*p* = 0.002). In multivariate analyses, strong CD3+ (HR: 0.163, CI: 0.044–0.614) and strong CD8+ TILs (HR: 0.265, CI: 0.081–0.864) were revealed as predictors for OS and the strong infiltration of CD20+ cells (HR: 0.095, CI: 0.019–0.464) for PFS. **Conclusions:** These first results of TILs in bladder SCC revealed predictive values of CD3+, CD8+ and CD20+.

## 1. Introduction

Europe has the highest incidence of bladder cancer (BC) with 11.6–36.7 cases per 100,000 inhabitants [1]. About 5% of cases are pure squamous cell carcinoma of the urinary bladder (SCC), of which 70% are muscle invasive (≥pT2) at initial diagnosis [2]. In African countries, schistosomiasis is the underlying trigger for the development of bladder SCC. In Western countries, chronic irritation and recurrent urinary tract infections are discussed as the main risk factors [1,2,3,4].

Compared with urothelial carcinoma (UC), bladder SCC is more frequently associated with advanced tumor stage and a worse prognosis [2]. Typically, bladder SCC patients have been excluded or underrepresented in clinical trials and, thus, little is known about treatment strategies [5]. So far, the results of neoadjuvant therapy are unsatisfactory and for high-risk and muscle-invasive bladder SCC, radical cystectomy (RC) remains the standard therapy [2,6]. 

The recurrence-free survival of BC patients undergoing RC is only 58–68% in the first 5 years, and numerous patients develop metastatic disease [7,8]. In metastatic stage, immune checkpoint inhibitors (ICIs) are an alternative treatment option to first-line platinum-based chemotherapy [6]. Encouraging study results also suggest the approval of ICI therapy in the adjuvant treatment of muscle-invasive bladder cancer (MIBC), as adjuvant ICI therapy after RC leads to significantly improved disease-free survival [9]. 

Despite all progress, the response rate to ICIs is reported to be only 20% and can be associated with severe side-effects and impaired quality of life [10,11,12,13,14]. New immunologic classification systems and an accurate characterization of the tumor microenvironment (TME) are needed to assess disease progression as accurately as possible and to better predict treatment response to ICIs [15,16,17,18]. In this context, tumor-infiltrating lymphocytes (TILs) as a specific component of the TME may be important, as TILs are a marker for the local antitumor immune response and a critical factor for successful ICI therapy [19,20,21,22].

The prognostic value of TILs has been previously demonstrated in numerous tumor entities. For example, a higher number of TILs correlates with better survival in cutaneous and mucosal melanoma and a high number of TILs is associated with improved prognosis in breast, colorectal, ovarian and pancreatic carcinomas [19,20,23,24,25,26,27]. In particular, it has been reported that a brisk infiltration of CD3+ and CD8+ cells is associated with a better oncological outcome. 

Study results also suggest that TILs may be useful markers for predicting survival in patients with MIBC and could help to identify patients who benefit most from ICI therapy [28,29,30,31,32,33,34]. In contrast, in variant histologic subtypes of bladder carcinoma, the prognostic value of TILs has not yet been well studied. In addition, most studies addressing TILs in BC do not list the histologic subtypes. However, since prognosis and therapeutic strategies differ, the differentiation of variant histologies could be critical for clinical practice [35]. 

While adenocarcinomas of the urinary bladder can be treated analogously to colorectal adenocarcinomas, there are no established chemotherapy options for bladder SCC [35]. Due to limited data, the treatment efficacy of ICIs in patients with advanced SCC of the urinary bladder is also widely unclear [5]. 

Analogous to UC, an improved understanding of tumor immunology, the TME and the prognostic impact of TILs could aid in selecting appropriate patients with bladder SCC for perioperative immunotherapy and facilitate treatment planning in this rare variant of BC. However, to date, studies on TILs in bladder SCC are lacking. In our study, we investigated subsets of TILs (CD3+, CD4+, CD8+, CD20+) and assessed their prognostic value for overall survival (OS) and progression-free-survival (PFS) in 61 patients with pure bladder SCC.

## 2. Material and Methods

### 2.1. Patients

The institutional ethics committee of Ludwig Maximilian University of Munich approved this study prior to initiation (Reference number 20–179). 

We retrospectively selected patients that had undergone RC due to MIBC between 2004 and 2019 at a high-volume academic center (Department of Urology, Ludwig-Maximilians-University, Munich). The patients were triaged for RC according to the published guidelines of the EAU: muscle-invasiveness, BCG-refractory non-muscle invasive bladder cancer and palliative reasons [6]. For patients who underwent RC in curative intent, a preoperative exclusion of distant metastasis was performed by computed tomography. Surgery was performed by trained urologists following standardized procedure, including pelvic lymphadenectomy and urinary diversion by ileal neobladder or ileal conduit. All the patients were offered post-operative rehabilitation and psycho-oncological care. A histological analysis was performed by experienced pathologists at the Department of Pathology, Ludwig Maximilians University, Munich. In 61 patients, the final histopathological specimen revealed pure bladder SCC. The specimens were reviewed to confirm the tumor type and stage, and staging was performed according to the AJCC/UICC TNM staging guidelines (8th edition) and the WHO classification of genitourinary tumors [36]. Cancer types that did not fulfil the histopathological criteria for pure SCC of the bladder as defined by the WHO 2016 system were excluded. Other exclusion criteria were low-grade tumors, neoadjuvant therapy, or a history of immunotherapy before RC. Patient characteristics, clinicopathological parameters, progression-free survival (PFS) and overall survival (OS) were documented and analyzed for correlation with intratumoral CD3+, CD4+, CD8+ and CD20+ lymphocyte infiltration.

### 2.2. Follow-Up

According to the EAU guidelines recommendation, frequent follow-up was carried out [6]. Additionally, patients were sent questionnaires regarding a report of the latest imaging twice in the first year after RC and then annually to update oncologic status. The inclusion criteria was written informed consent, and the data collection followed the World Medical Association and Declaration of Helsinki.

### 2.3. TMA Construction and Immunohistological Analysis 

TILs (CD3+, CD4+, CD8+, CD20+) were examined by immunohistochemistry on tissue microarrays (TMAs) containing 171 tumor tissue cores from 61 patients with bladder SCC.

TMAs were generated using all available formalin-fixed and paraffin-embedded (FFPE) tumor tissue blocks. Three different tumor regions were identified and marked on hematoxylin and eosin (H&E) stained sections that were matched to their corresponding paraffin blocks (donor blocks). Three tumor cores with a diameter of 1 mm were punched from these donor blocks and precisely arrayed into a new recipient paraffin block (TMA block). Each sample was arrayed in triplicate cores to minimize tissue loss and to overcome tumor heterogeneity. 

Slices (4 μm) were cut from the TMA blocks and transferred to slides, which were deparaffinated, rehydrated and automatically stained using Ventana BenchMark XT (Ventana Medical Systems, Roche, Basel, Switzerland), according to the standard operating procedure. Antigen retrieval was conducted with CC1 (Ventana) for 64 min (CD3, CD4, CD8) or 36 min, respectively (CD20).

The following antibodies were used for the staining of TMA sections: anti-CD3 monoclonal rabbit (SP7, Zytomed Systems, Berlin, Germany) at 1:150 dilution for 28 min; anti-CD4 mouse monoclonal (4B12, Leica Biosystems, Wetzlar, Germany) at 1:500 dilution for 32 min; anti-CD8 mouse monoclonal (C8/144B, Medac, Wedel, Germany) at 1:50 dilution for 24 min; and anti-CD20 mouse monoclonal (L26, Dako Agilent, Santa Clara, CA, USA) at 1:400 dilution for 20 min. Ventanas UltraView (CD3, CD8 and CD20) and OptiView (CD4) Universal DAB Detection Kits were used to detect protein expression. Tonsil tissue was used as a positive control for each biomarker and was included in each staining run.

### 2.4. Semiquantitative Analysis of TILs

The expression of CD3, CD4, CD8, and CD20 in TILs was examined by immunohistochemistry on TMAs, as previously described [37]. For each staining, the number of immunoreactive cells in each TMA core was manually counted at ×400 magnification (×40 objective). The mean of the readings in the three related tumor cores was then calculated and represented the number of positive TIL subtypes per tumor sample. If a TMA core was lost or did not contain tumor tissue, this core was excluded for the overall evaluation. The respective median value of immunoreactive cells was used as a cutoff value for subdivision into weak and strong lymphocyte infiltration (CD3 (≥273); CD4 (≥4); CD8 (≥106), CD20 (≥7)). Examples of the different TIL groups are shown in Figure 1.

### 2.5. Statistics

The primary endpoints were PFS and OS. OS was defined as the time between surgery and death. PFS refers to the time between primary surgery and disease recurrence. Disease recurrence was defined as the occurrence of locoregional recurrence or distant metastasis on radiological imaging diagnostics. Patients who were alive at the end of follow-up were censored. Survival curves were calculated using the Kaplan–Meier method and compared by a log-rank test. Associations between TIL grades and patient characteristics were analyzed using a Mann–Whitney and a chi-square test. To assess the prognostic values of TIL grades on survival parameters, multivariate analyses were performed using Cox regression models adjusted for age, gender, TNM stage and grading. All calculations were performed using SPSS version 28.0 (IBM, Armonk, NY, USA) and MedCalc version 20.106 (Ostend, Belgium). *p*-values smaller than 0.05 were considered statistically significant. 

## 3. Results

### 3.1. Demographics and Clinical Characteristics

Out of 1600 patients undergoing RC between 2004 and 2019, 61 patients revealed pure SCC of the bladder in the final histological specimen. The median age of all patients at the time of RC was 66 (range: 37–86) years and the median follow-up of all patients was 16 (range 0–175) months. A total of 28 patients (45.9%) died, and 26 patients (42.6%) revealed recurrence or distant metastasis during follow-up. A summary of patient characteristics is given in Appendix A.

### 3.2. Associations between TILs and Clinicopathological Characteristics

A strong lymphocyte infiltration was seen in 27 tumors for CD3+ (44%); in 28 tumors for CD4+ (46%); in 26 tumors for CD8+ (43%); and in 27 tumors for CD20+ lymphocytes (44%). A total of 12 tumors showed a strong infiltration with all four lymphocyte subgroups (20%), whereas 20 tumors showed a weak infiltration for all four subgroups (33%). 

There were no significant relationships between TIL scores (CD3+, CD4+, CD8+, CD20+) and clinical (age: *p* = 0.591–0.920; sex: *p* = 0.531–0.913) or histopathological characteristics (pT: *p* = 0.139–0.580; pN: *p* = 0.082–0.693; M stage: *p* = 0.258–0.902; grading: *p* = 0.127–0.973). Moreover, there were no correlations between TIL scores and resection margins (CD3 *p* = 0.831; CD4 *p* = 0.510; CD8 *p* = 0.815; CD20 *p* = 0.330). 

### 3.3. TILs and Overall Survival

The median OS of all patients was 21 months (95% CI: 14–87 months). Significantly improved 2-year OS was observed for tumors with strong CD3+ (2-year OS: 72.4% vs. 21.7%, *p* < 0.001); CD4+ (2-year OS: 56.7% vs. 36.9%, *p* = 0.045); CD8+ (2-year OS: 68.9% vs. 30.2%, *p* = 0.001); and CD20+ (2-year OS: 64.5% vs. 30.4%, *p* < 0.001) lymphocytic infiltration, as compared to tumors with weak lymphocytic infiltration. (Figure 2). 

### 3.4. TILs and Progression-Free Survival

The median PFS of all patients was 16 months (95% CI: 7–17 months). A strong tumor-infiltration for CD3+ (2-year PFS: 66.3% vs. 29.6%, *p* = 0.025) and CD20+ (2-year PFS: 68.5% vs. 22.6%, *p* = 0.002) was associated with a longer PFS, as compared to patients with weak CD3+ and CD20+ infiltration. The differences between subgroups with strong and weak infiltration of CD4+ (2-year PFS: 55.7% vs. 39.1%, *p* = 0.292) and CD8+ (2-year PFS: 64.5% vs. 33.5%, *p* = 0.069) were not statistically significant (Figure 2).

### 3.5. Multivariate Analyses

Multivariable analyses were calculated for OS, CSS and PFS. Strong infiltration of CD3+ (HR: 0.163, CI: 0.044–0.614) and CD8+ (HR: 0.265, CI: 0.081–0.864) were identified as independent predictors for improved OS and CSS. A positive nodal status (HR: 14.178, CI: 3.075–65.376) and high-grade pattern (HR: 5.189, CI: 1.288–20.901) were independent predictors for shorter OS. In addition, a strong infiltration with CD20+ (HR: 0.095, CI: 0.019–0.464) was an independent predictor for longer PFS. (Figure 3).

## 4. Discussion

Bladder SCCs account for approximately 3–5% of all BC, making them the most common non-urothelial subtype of BC [2,3]. Since bladder SCC patients are often underrepresented in clinical trials, little is known about the treatment efficacy and use of ICIs in pure bladder SCC [5]. In addition, studies on the composition of TILs in this tumor entity are lacking so far. Given the poor prognosis and limited treatment options in advanced bladder SCC, there is an unmet need for reliable new prognostic markers and immunologic classification systems that could help in the planning of treatment and follow-up [15,16,17,18].

Here, we investigated TILs in the RC specimen from patients with bladder SCC, in order to improve the understanding of the immunologic tumor milieu in this cancer entity and its impact on PFS and OS. Consistent with the literature, 61 of 1600 patients (3, 8%) in our cohort showed SCC. We found that strong infiltrates of CD3+, CD4+, CD8+, and CD20+ lymphocytes are associated with prolonged OS, as patients with strong intratumoral lymphocytic infiltrates survived significantly longer than patients with weak TILs. CD3+ and CD8+ TILs also turned out to be independent predictors of improved OS in the multivariate analysis. In addition, high numbers of CD3+ and CD20+ TILs were significantly associated with prolonged PFS, and for CD20+ TILs, this significance remained in a multivariable model. Our results suggest that TILs are a reliable prognostic marker in patients who undergo RC for bladder SCC.

The interaction between tumor cells and the immune system plays a crucial role in the development and progression of cancer. Activated TILs can mediate an antitumor immune response that ultimately leads to tumor cell death [38,39]. In BC, antitumor effects of the immune system have been used for many years in the context of intravesical therapy with bacillus guermette guerin (BCG) [40]. Recently, the antitumor immune response in BC has been additionally utilized for therapeutic purposes by the use of ICIs, which aim to counteract tumor-mediated immunosuppression [41]. The drugs used in the therapy of BC, pembrolizumab and atezolizumab, represent monoclonal antibodies targeting the checkpoint protein PD-1 (programmed cell death protein 1) and its ligand PD-L1 [12,14]. Unfortunately, in large clinical trials only about 20% of BC patients benefited from ICI therapy [10,11,12,13,14].

The successful response to immunotherapies depends on the immunological composition of the TME [42]. Immunogenic tumors with strong infiltration by activated immune cells are often associated with a better therapeutic response to ICI therapy [43,44]. Therefore, an assessment of TILs could help to identify patients who would benefit most from ICI therapy [21,22,30].

Among TILs, CD3+ T cells and cytotoxic CD8+ lymphocytes are essential for the antitumor immune response [32,42,43,45]. Consequently, numerous studies have investigated the prognostic value of CD3+ and CD8+ TILs in UC. Some studies showed that high TILs correlated with a worse outcome [46,47,48]. However, most studies identified a high density of CD3+ and CD8+ lymphocytes as a positive prognostic marker. For example, in non-muscle invasive bladder cancer (NIMBC) low CD3+ TILs and low CD8+ TILs were associated with a high risk of relapse and high TILs were related with a prolonged OS [49,50]. In stroma-invasive high-grade BC, strong infiltration of the tumor stroma by CD3+ lymphocytes was associated with a favorable outcome [51]. In MIBC, a high density of CD3+ and CD8+ cells in the tumor margins of MIBC was associated with a better oncologic outcome after RC and high densities of CD3+ and CD8+ lymphocytes were associated with better OS and disease-free survival [29,31,32,52]. 

These findings are in line with our results in bladder SCC. Here, strong infiltrates of CD3+ and CD8+ lymphocytes were independent predictors of improved OS. In addition, high CD3+ TILs were significantly associated with a prolonged PFS. We therefore identified CD3+ and CD8+ TILs as a reliable prognostic tool in patients with bladder SCC. Interestingly, in our study, 43% of all bladder SCCs showed strong infiltration of CD8+ cells and 44% were strongly infiltrated by CD3+ cells. Utilizing this high rate of TILs promises great potential for immune-based therapies in advanced SCC, eventually even in an adjuvant setting.

CD4+ cells are involved in heterogeneous functions of the immune system and play a central role in modulating the immune response [53]. Although CD4+ cells are not the target of anti-PD-1/PD-L1 therapies, these cells still have a critical impact on the effectiveness of the antitumor immune response through their regulatory functions, and the co-occurrence and interaction of CD4+ and CD8+ lymphocytes represents an important mechanism in tumor detection and clearance [53,54,55]. In BC, only preliminary data on the role of CD4+ TILs in the antitumor immune response are available. In one study, Oh et al. found that CD4+ lymphocytes mediate anti-tumor cytotoxicity [54]. In our study, there was a significant correlation between strong CD4+ TIL infiltration and improved OS in patients with bladder SCC. However, in the multivariate analysis, CD4+ lymphocytes were no independent predictor for OS or PFS. Accordingly, the specific impact of CD4+ in bladder SCC remains uncertain, and further studies on CD4+ lymphocytes and their subtypes are needed to characterize the antitumor function of CD4+ TILs in bladder SCC. 

CD20+ TILs have not been well studied in trials to date, and differences in B-cell related antitumor-functions were found between different tumor entities [56]. Thus, the prognostic value of CD20+ B cells in terms of oncologic outcome and response to immunotherapy remains controversial. Subgroups of B cells provide an antibody-mediated immune response to tumor antigens. In addition, CD20+ cells may contribute to the enhancement of cytotoxic antitumor effects via the activation of T cells [56,57]. In MIBC, one study found that follicular structures with CD20+ TILs were associated with longer survival after RC [58]. These results are consistent with the findings of our study. We found significantly prolonged OS and PFS in patients with a strong intratumoral CD20+ infiltration. The significant association between strong CD20+ TILs and improved PFS was also maintained in multivariable analyses. Thus, our results suggest that CD20+ TILs have an inhibitory effect on disease progression and are an independent predictor of improved PFS.

A limitation of our study is the retrospective examination of data at a single center. In addition, the number of cases remains comparatively small due to the rarity of pure bladder SCC. Due to the limited number of cases, a sub-analysis of different tumor stages (T2 vs. T3 vs. T4) or N stages (N0 vs. N1 vs. N2) was not performed. This should be the subject of further multi-center studies with larger case numbers. Additionally, due to the large study period (2004–2019), patients received different regimens of neoadjuvant therapy, which could influence oncological outcome. Nevertheless, we provide the first overview of TILs in bladder SCC. CD3+, CD8+, and CD20+ TILs are strong independent predictors of improved OS and PFS, respectively, and may help stratify patients for adjuvant ICI therapy of bladder SCC after RC. Further prospective (multicenter) studies are needed to externally validate our findings.

## 5. Conclusions

Here, we present the first results on tumor-infiltrating lymphocytes (TILs) in pure squamous cell carcinoma of the urinary bladder. Strong tumor infiltration with CD3+, CD4+, CD8+, and CD20+ immune cells correlated significantly with improved OS. In the multivariate analysis, high CD3+ and CD8+ TILs were revealed as independent predictors of enhanced OS. In addition, strong CD20+ infiltration was an independent predictor of prolonged PFS. Our results suggest that the evaluation of TILs is an important prognostic tool in patients with bladder SCC. Further research is needed to confirm our findings.

## Figures and Tables

**Figure 1 cancers-14-03999-f001:**
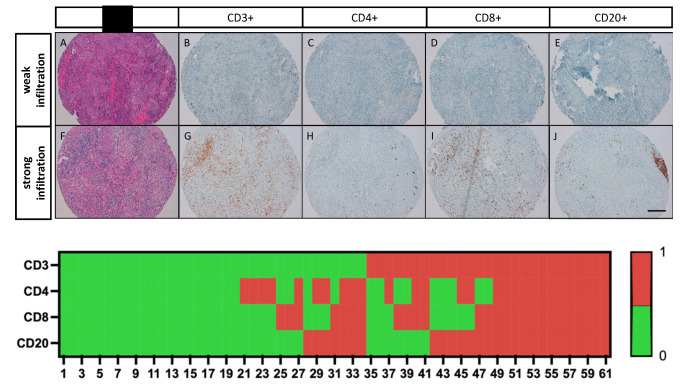
Immunoreactivity of TIL subtypes in bladder SCC and case-related expression patterns of CD3, CD4, CD8 and CD20. A–E representative photomicrographs show the same TMA sample with pure bladder SCC (**A**,**H**,**E**) and weak infiltration with CD3+ (**B**); CD4+ (**C**); CD8+ (**D**); and CD20+ TILs (**E**). (**F**–**J**) representative photomicrographs show the same TMA sample with pure bladder SCC (**F**,**H**,**E**) and strong infiltration with CD3+ (**G**); CD4+ (**H**); CD8+ (**I**); and CD20+ TILs (**J**). Scale bar = 100 µm.

**Figure 2 cancers-14-03999-f002:**
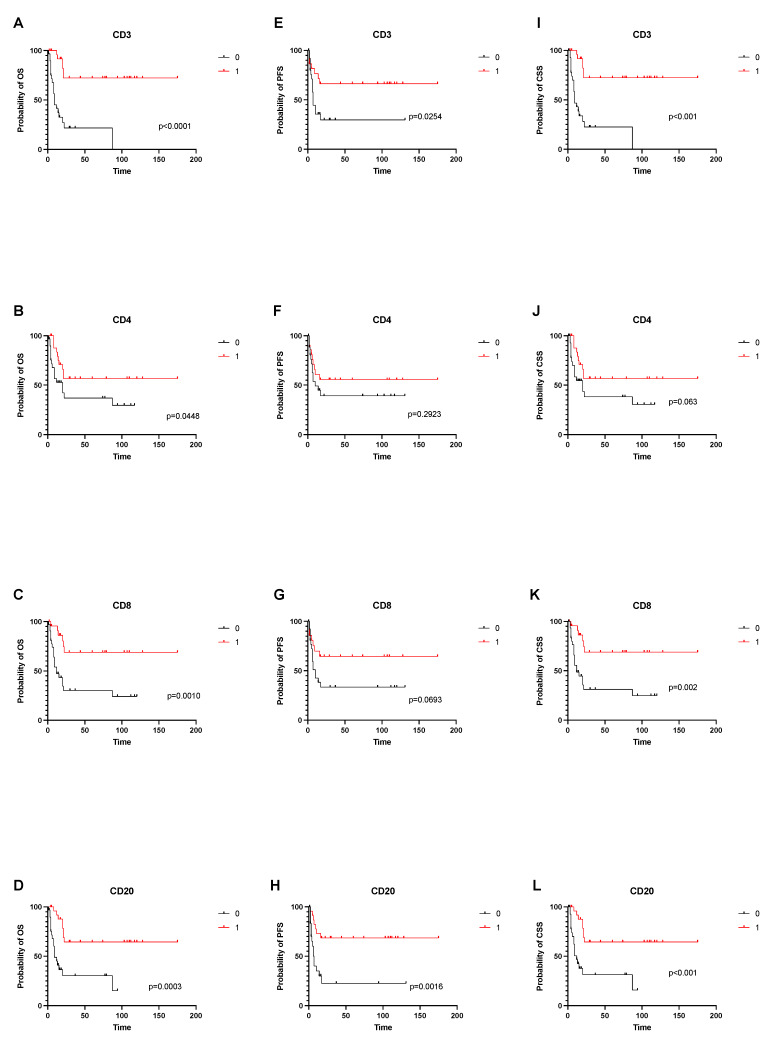
Overall, cancer-specific and progression free survival depending on tumor-infiltrating immune cells. (**A**–**D**) shows OS depending on infiltration status of CD3, CD4, CD8 and CD20 cells. (**E**–**H**) shows PFS depending on infiltrations status of CD3, CD4, CD8 and CD20 cells. (**I**–**L**) shows CSS depending on infiltrations status of CD3, CD4, CD8 and CD20 cells.

**Figure 3 cancers-14-03999-f003:**
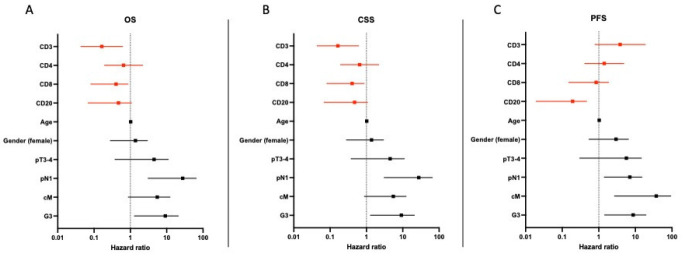
Multivariate analysis of predictors of overall, cancer-specific and progression-free survival in patients with pure SCC of the bladder. (**A**–**C**) shows hazard ratio for tumor specific parameters and its impact on OS, CSS and PFS.

## Data Availability

All data generated or analyzed during this study are included in this manuscript. Further enquiries can be directed to the corresponding author.

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
