# Peer review of "Identification of the Tumor Infiltrating Lymphocytes (TILs) Landscape in Pure Squamous Cell Carcinoma of the Bladder"

_cancers, 2022, doi:10.3390/cancers14163999_

Round 1

Reviewer 1 Report

Strengths

·      Multiple tissue cores from each sample

·      Use progression free survival as an endpoint as well as overall survival

·      Multivariate analysis used to adjust for age, gender, TNM stage, and grade

To be addressed:

Introduction

·      Could you include statistics about the prevalence of SCC within MIBC overall? And then compare this percentage to the percentage in your population sample (61/1600)?

·      Line 58: “…and TILs are associated with improved prognosis in breast, colo-rectal, ovarian and pancreatic carcinomas.” Does this mean a higher number of TILs? More variation of TILs? Please specify.

Methods

·      Can you include a line explaining how the cut-off was determined which separated samples into “strong” and “weak” infiltration?

Results/Discussion

·      Positive nodal status and grade were also independent predictors. Can the analysis be run excluding the 15 positive nodal samples to see if the significance of the TILs remains? Same for the tumor grade. Without these analyses there should be discussiona bout how this affects the results.

·      There is no tumor grade data in Supplementary table 1

Reviewer 2 Report

The manuscript by Eismann et.al. retrospectively investigated relationship between different subsets of TILs and survival outcomes in a cohort of patients with pure SCC of bladder. Several issues need to be addressed before the manuscript is considered for publication.

1. Please check/revise Figure 1 and Figure 3 - they do not display correctly in the current submitted version.

2. Please provide a clear definition of progression free survival (PFS)

3. Did any patients in the cohort receive adjuvant therapy, including immune checkpoint inhibitors? 

Reviewer 3 Report

The authors present a study on tumor-infiltrating lymphocytes (TILs) and their impact on OS in squamous cell carcinoma of the bladder. It covers a very interesting topic in cancer diagnostics and is higly related also the potential effects on outcome and treatment. They present a novel possibility for tumour  subclassification which likely could have an impact on treatment response in case of adjuvant therapy. Therefore it fits the journals perspectives and recommendations. It derives from an outstanding large cohort from over 1600 cases and might be one of the largest studies looking at the squamous subtype. The multivariate analysis shows a significant correlation with TILs and OS. This finding is new and important.  The authors should be congratulated for their idea, work and effort they have put into performing this study. It is well written (English good). Abstract good. Stats good. References good. Figures and tables good.

Suggestion to the editor: accept with minor revisions.

However, some sections of the manuscript need to clarified:

Title:

This comment might be more a personal impression/suggestion ( in fact for title and manuscript): I strongly believe that the study is of highest value. Maybe the focus should be more on “identification of the TILs landscape in pure squamous cell carcinoma”  than OS. The finding alone is important enough. Direct correlation with OS might have a bias (see later).  Please comment.

Results / Discussion:

Limitations should be mentioned in the limitations section of the document:

-          Stratification for T Stage, N stage, Tumor size: would a sub analysis of T2 vs. T3 vs T4 ad value to you study. Or N-/+. Please comment.

-          Did you look at R0 vs. R1 (R2)?

-          Endpoint CSS is more common. Please comment why you have chosen not to report CSS

-          Endpoint OS: patient age range from 37-86… please comment

-          Did some of these patients receive neoadjuvant or adjuvant therapies? The cohort covers many years. In that time systemic treatment strategies have changed. Please comment.

Suppl Table 1: last column seems to low (graphically speaking)

Finally:

I encourage the authors to look at cases with non-muscle invasive tumours. Is there a correlation with TILs and later recurrence or progression? I there is a correlation this could potentially be used as a predictor an could help selecting patients for early cystectomy.

Round 2

Reviewer 2 Report

Thank you for addressing the comments. It looks good to me. No more additional comments.